# A 102 W High-Power Linearly-Polarized All-Fiber Single-Frequency Laser at 1560 nm

Jiamin Huang [1], Qilai Zhao [2,3,4,5,*], Junjie Zheng [2,3,4,5], Chengzi Huang [2,3,4,5], Quan Gu [1,3,4,5], Wanpeng Jiang [2,3,4,5], Kaijun Zhou [3,4,5], Changsheng Yang [3,4,5,6,7], Zhouming Feng [3,4,5,6,7], Qinyuan Zhang [2,3,4,5], Zhongmin Yang [2,3,4,5] and Shanhui Xu [2,3,4,5,6,7,*]

1   School of Materials of Science and Engineering, South China University of Technology, Guangzhou 510640, China; jmhuang0925@foxmail.com (J.H.); 201921021032@mail.scut.edu.cn (Q.G.)
2   School of Physics and Optoelectronics, South China University of Technology, Guangzhou 510640, China; 201765503021@mail.scut.edu.cn (J.Z.); 201920128737@mail.scut.edu.cn (C.H.); 202020130464@mail.scut.edu.cn (W.J.); qyzhang@scut.edu.cn (Q.Z.); yangzm@scut.edu.cn (Z.Y.)
3   State Key Laboratory of Luminescent Materials and Devices and Institute of Optical Communication Materials, South China University of Technology, Guangzhou 510640, China; stephenzhou@lightpromotech.com (K.Z.); mscsyang@scut.edu.cn (C.Y.); zhmfeng@scut.edu.cn (Z.F.)
4   Guangdong Engineering Technology Research and Development Center of Special Optical Fiber Materials and Devices, South China University of Technology, Guangzhou 510640, China
5   Guangdong Provincial Key Laboratory of Fiber laser Materials and Applied Techniques, South China University of Technology, Guangzhou 510640, China
6   Guangdong Engineering Technology Research and Development Center of High-Performance Fiber Laser Techniques and Equipments, Zhuhai 519031, China
7   Hengqin Firay Sci-Tech Company Ltd., Zhuhai 519031, China
*   Correspondence: zhaoql@scut.edu.cn (Q.Z.); flxshy@scut.edu.cn (S.X.)

**Abstract:** A 1560 nm high-power linearly-polarized all-fiber single-frequency narrow-linewidth laser with near diffraction-limited beam quality is demonstrated. The Yb–Er energy transfer efficiency and the ability of the signal laser to capture pump light have been improved by specifically choosing the pumping wavelength and the input signal power in the final power amplifier stage of this laser system. Under the off-peak absorption pumping wavelength of 940 nm, along with the maximum input signal power of 6 W, a maximum output power of 102 W with a slope efficiency of 40.5% is acquired. At the highest output power status, a polarization extinction ratio (PER) of 15.5 dB, a linewidth of 3.05 kHz, and a beam quality of $M_x^2 = 1.14$, $M_y^2 = 1.06$ are obtained, respectively. This advanced single-frequency fiber laser has great potential for the long-range coherent Doppler lidar and the next generation of gravitational wave detection.

**Keywords:** single frequency; fiber laser; high power; linearly-polarized

## 1. Introduction

1.5 μm single-frequency fiber lasers (SFFLs) have the noteworthy advantages of high transmission performance in the atmosphere, high damage threshold to the human eye, and long coherence distance [1,2], which can be widely used in coherent optical communication [3], laser biomedical technology [4], and nonlinear frequency conversion [5]. With regard to the coherent Doppler lidar, the distance improvement of space target velocity measurement heavily depends on optimizing its characteristics, such as the high output power and narrow linewidth of the 1.5 μm SFFL [6,7]. In particular, a high-power 1.5 μm SFFL with linearly-polarized radiation and excellent beam quality is proposed as a potential source for next-generation gravitational wave detectors (GWDs), since silicon, which will be used as a material for optical substrates due to its excellent properties at cryogenic temperatures, is not transparent at 1064 nm [8–10]. The above requirements are

extremely challenging and unique, so the design and demonstration of this state-of-the-art 1.5 μm SFFL has become a crucial task in the field of fiber lasers.

We know that the 1.5 μm SFFLs, relying on a single oscillator, can only obtain a power of a few hundred milliwatts [11], so a master oscillator power amplifier (MOPA) is indispensable to further enhance the output power of SFFLs [12,13]. However, the conventional fiber-based MOPA systems, especially applied to SFFL, are prone to the stimulated Brillouin scattering (SBS) effect, due to their small core diameter and high energy density. The SBS effect not only limits the further increase in power, but also easily causes instability in the laser setup [14]. Increasing the effective mode field area of the fiber and shortening the fiber length can effectively improve the SBS threshold power, so large mode area (LMA) fibers are widely used in MOPA systems [15]. In previous research, an output power of 207 W was realized by utilizing LMA Er–Yb co-doped fiber, which is the highest output power of 1.5 μm single-frequency fiber amplifiers [16]. However, using this utilized non-polarization-maintaining (nPM) format, it is difficult to acquire a stable linearly-polarized beam, resulting in application limitations to coherent detection. In 2017, a 1.5 μm SFFL with an output power of 110 W and a polarization extinction ratio (PER) of 13 dB was reported using polarization-maintaining large-mode-area (PLMA) active fiber [8]. Nonetheless, the utilized spatial components made the laser sensitive to environmental disturbances and coupled fluctuations, which will degrade the laser noise characteristics and is contrary to the requirement of the GWDs. Recently, a fully monolithic single-frequency Er–Yb co-doped fiber amplifier, with an all-fiber structure at 1556 nm, was displayed to provide 110 W output power, whereas the PER was only at the 10 dB level, and the linewidth was up to MHz order of magnitude [17]. So far, it remains a huge challenge to achieve a high-power linearly-polarized all-fiber laser output, particularly for narrow-linewidth single-frequency radiation.

In this article, we have demonstrated a 1560 nm high-power linearly-polarized SFFL based on the optimized cascaded MOPA structures. The effect of pump wavelengths and the input signal power on the output performance of the power amplifier is investigated. By using the 940-nm off-peak pumping scheme and inputting 6 W signal power, an output power of 102 W, a slope efficiency of 40.5%, a PER of 15.5 dB, and a linewidth of 3.05 kHz is obtained. Moreover, the use of a 25 μm core diameter PLMA Er–Yb co-doped fiber has lifted the limitation of the SBS effect. Then, by virtue of a racetrack-shapep coiling, with a minimum inner diameter of 8 cm, a near diffraction-limited beam quality of $M_x^2 = 1.14$, $M_y^2 = 1.06$ is realized.

## 2. Experimental Setup

The schematic of the 1560 nm high-power linearly-polarized SFFL is shown in Figure 1. The laser system consists of a seed oscillator, two pre-amplifiers, and a power amplifier. The seed oscillator is formed by a polarization-maintaining (PM) narrow-band fiber Bragg grating (FBG) and a high-reflected wide-band fiber Bragg grating (HR-FBG) on each end of a homemade 1.8 cm long high-concentration Er–Yb co-doped phosphate fiber (EYPF). More details of the fiber properties and the seed oscillator can be found in our previous works [11,18]. Utilizing a 976 nm single-mode laser diode (SM-LD) counter-pumping through a 980/1560 PM wavelength division multiplexer (PM-WDM), a laser beam with a power of 30 mW, a linewidth of 3.05 kHz, and a PER of 22 dB is output from a PM-isolator (PM-ISO).

The two pre-amplifiers have applied the same counter-pumping structure via a PM signal/pump (2 + 1) × 1 combiner. The active medium in the pre-amplifiers is a PM Er–Yb co-doped double-cladding fiber (EYDF) with a core/inner-cladding diameter of 10/128 μm and an absorption coefficient of 2 dB/m @ 915 nm. To synthetically manage the effects of amplified spontaneous emission (ASE) and SBS, the length of the active fiber and the pump wavelength in the two-stage pre-amplifiers are optimized. In the first pre-amplifier, due to the low input signal optical power, the lack of gain competition ability will produce a large amount of ASE. Therefore, an off-peak absorbing 940 nm MM-LD is utilized to reduce the

number of spontaneously emitted particles. The absorption coefficient of the gain fiber is 2.4 dB/m@940 nm, using a 5.2 m long gain fiber to make the total absorption coefficient reach 12.5 dB. In the second pre-amplifier, the effect of the SBS on further power growth cannot be ignored. Under the comprehensive consideration of the SBS threshold and total absorption coefficient, the length of the active fiber is shortened to 2.4 m, while the 976 nm wavelength pump is finally adaptively selected to achieve a 15 dB absorption coefficient. A composite device (PMTI-WDM), including the functions of a tap-end isolator and a WDM, is introduced to monitor the output light performance and filter out the backward Yb-ASE produced by the fiber amplifier. After the two pre-amplifiers, the power of the signal laser can increase up to 6 W.

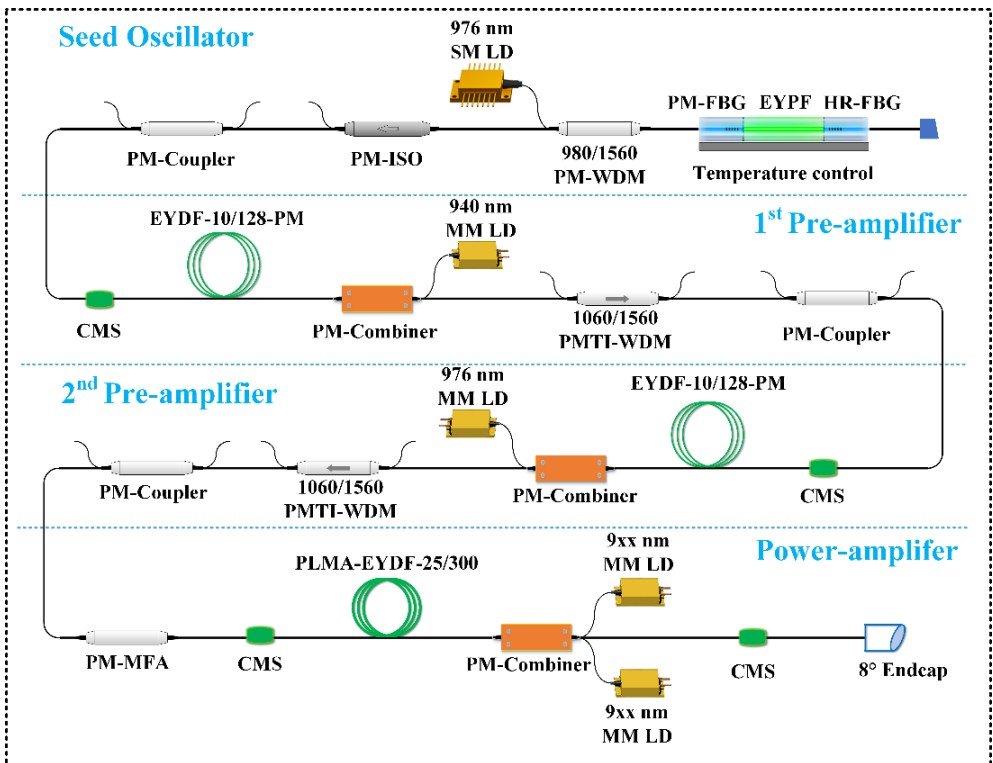

**Figure 1.** Experimental setup of the 1560 nm high-power linearly-polarized SFFL. (HR-FBG: high-reflected fiber Bragg grating; EYPF: Er–Yb co-doped phosphate fiber; PM: polarization-maintaining; WDM: wavelength division multiplexer; SM: single-mode; LD: laser diode; ISO: isolator; CMS: cladding-mode stripper; EYDF: Er–Yb co-doped double-cladding fiber; MM: multi-mode; TI: tap isolator; MFA: mode field adaptor; PLMA: PM large-mode-area).

The power amplifier employs a PLMA active fiber, with a core/inner-cladding diameter of 25/300 μm, to improve the threshold of the SBS effect. The length of the PLMA-EYDF is 5.5 m, the cladding absorption of which is 3 dB/m@915 nm. Two 9xx nm (940 nm or 976 nm) MM-LD are injected into the active fiber through a PM-combiner, which also adopts the method of counter-pumping to enhance the output power. A PM mode field adaptor (PM-MFA) delivers a transition link method of reducing the fusion loss of the PLMA fiber and the PM-1550 fiber. The output end face of the PM-combiner is angle-cleaved by ~8 degrees for eliminating Fresnel reflections. The PLMA fiber, with a larger core, can transmit multiple modes of transverse mode light. To filter out the generated high-order modes during the propagating process, the PLMA-EYDF is coiled into a race-track shape with a minimum inner diameter and a maximum outer diameter of 8 cm and 14 cm, respectively.

Furthermore, along with the second pre-amplifier and power amplifier, the optical fiber device, pump source, and active fiber of the second pre-amplifier are plastered to

water-cooling aluminum plates with a set temperature of 21 °C for efficient heat control. Considering the large heat accumulation, the active fiber of the power amplifier stage is directly placed into the groove of the water-cooling plate surface via water immersing. To achieve better thermal conduction, the fusion point between the optical fiber devices is closely attached to the water-cooled plate, especially for the fusion positions between the second pre-amplifier and main amplifier. Thanks to the optimization of fiber splice parameters, there is no catastrophic damage at the fusion point. In order to ensure the reliability and stability of the splice point, the angle deviation of the splice point is controlled to be less than 0.1° and the loss to be less than 0.02 dB. To realize a safe operation, the temperature of the system is carefully monitored during the process of increasing the pump power. There is no abnormal thermal increase at each splice point. At the maximum output power situation, the temperature of the active fiber immersed in water is about 28 °C, and that of other devices and fusion points remains at 30 °C. Four cladding-mode strippers (CMSs) are employed to remove the residual pump laser and high-order mode from the fiber cladding in this laser system. Three 2 × 2 PM couplers are inserted to synchronously monitor the forward and reverse laser characteristics. The PM-ISO and associated devices are spliced to prevent damage from the backward propagating light.

## 3. Results and Discussion

Different pumping wavelengths and input signal powers in the power amplifier are utilized to investigate the impact on the laser output characteristics, respectively. First, with an input signal power of 4.9 W, the output power versus the pump power of 940 nm and 976 nm LD are explored, as shown in Figure 2a, and the corresponding spectrum with the same 1560 nm output power of 40 W is given in Figure 2b. The 940 nm LD, with an off-peak absorption pumping, has a higher slope efficiency of 38.9% compared to a slope efficiency of 32.1% using the 976 nm LD. This phenomenon can be attributed to the fact that the off-peak absorption pumping method improves the Yb–Er energy transfer efficiency, thereby increasing the threshold power of Yb-ASE, allowing more pump light to be used for signal power enhancement [8]. Concerning the optical spectrum, obvious ASE near 1535 nm can be found under 976 nm LD pumping, which deteriorates the optical signal-to-noise ratio (OSNR) of the output laser. By contrast, with 940 nm LD pumping, a superior optical spectrum with an OSNR of 64 dB has been displayed. The reason for this phenomenon can be attributed to the fact that to obtain the same output optical power, more pump laser needs to be injected under the 976 nm LD pumping condition, resulting in some superfluous $Er^{3+}$ particles of upper energy levels to experience spontaneous emission.

Figure 2c shows the energy level diagrams in the Er–Yb co-doped fiber system. Under the strong absorption of 976 nm pumping, a large number of $Yb^{3+}$ particles are excited to the $^2F_{5/2}$ energy stage, transferring the energy to the acceptor $Er^{3+}$ particles through cross-relaxation [8,17]. A number of $Yb^{3+}$ particles located at the $^2F_{5/2}$ stage will generate Yb-ASE through nonradiative transitions. Similarly, when a large number of $Er^{3+}$ particles are located in the excited state $^4I_{11/2}$, the redundant particles will also relax to the ground state through spontaneous radiation, resulting in the 1.5 μm ASE. Under 940 nm off-peak pumping, the absorption cross-section of $Yb^{3+}$ particles to pump light is smaller. Therefore, the number of $Yb^{3+}$–$Er^{3+}$ particles located at the excited stage is reduced, with the ASE in the 1.0 μm and 1.5 μm bands effectively suppressed. Therefore, adopting an off-peak pumping scheme can effectively increase the output power and optimize the spectrum of the laser.

Second, with various input signal powers of 2.7 W, 4.9 W, and 6 W, the output power versus the pump power of 940 nm LD is plotted in Figure 3a. It can be noted that the slope efficiency of the output laser has increased from 37.2% to 42.9% as the input signal power gradually enhances. With the boost of the input signal power, the capture ability of the signal laser for pump light has improved, contributing to the promotion of slope efficiency. Figure 3b demonstrates the laser spectrum results with the same output power of 40 W from different input signal powers. Compared with the condition of high injection signal

power, the ASE component of the output laser is slightly more obvious at a low injection power. The origin of this consequence is that the pickup capability of the signal particle is inadequate during the low power of the injected signal laser. Consequently, some particles located in the $Er^{3+}$ metastable energy level fail to generate stimulated emission, which directly experiences spontaneous emission. Fortunately, the OSNR of the spectrum is still as high as 64 dB at different input powers.

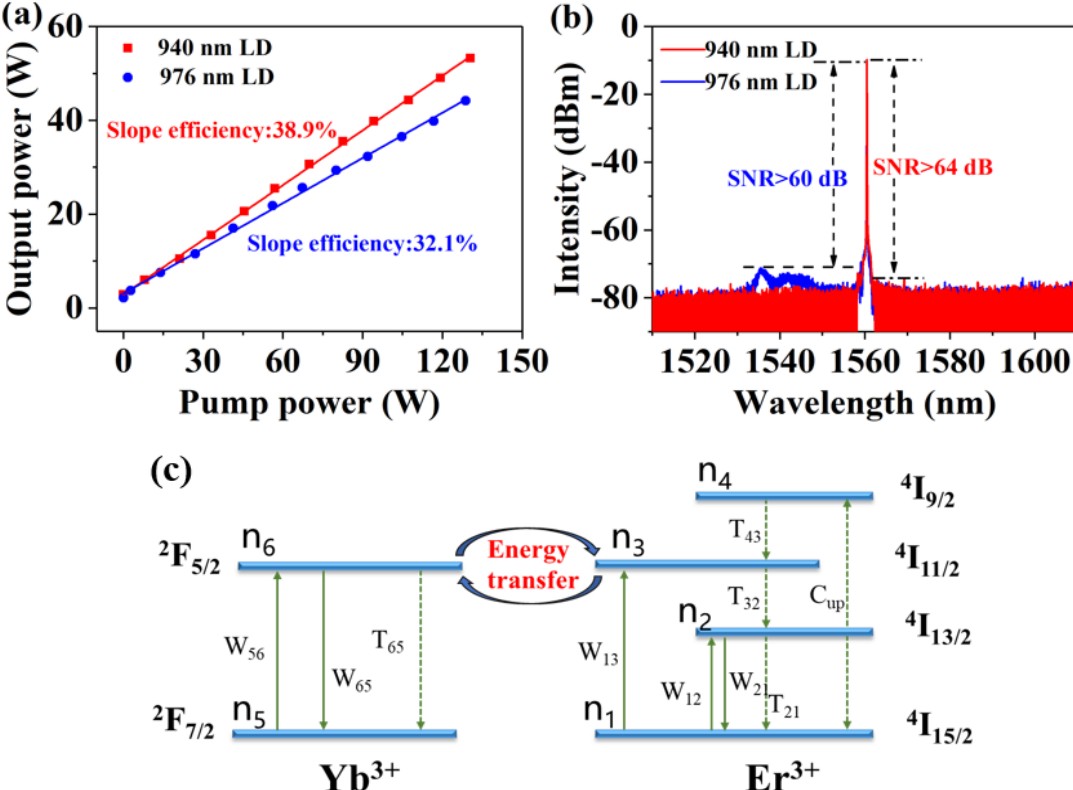

**Figure 2.** The system is under different pumping wavelengths. (**a**) The output power of 1560 nm linearly-polarized SFFL versus the pump power. (**b**) The output spectrum of 1560 nm linearly-polarized SFFL under 40 W output power. (**c**) Energy level diagrams in Er–Yb co-doped fiber system.

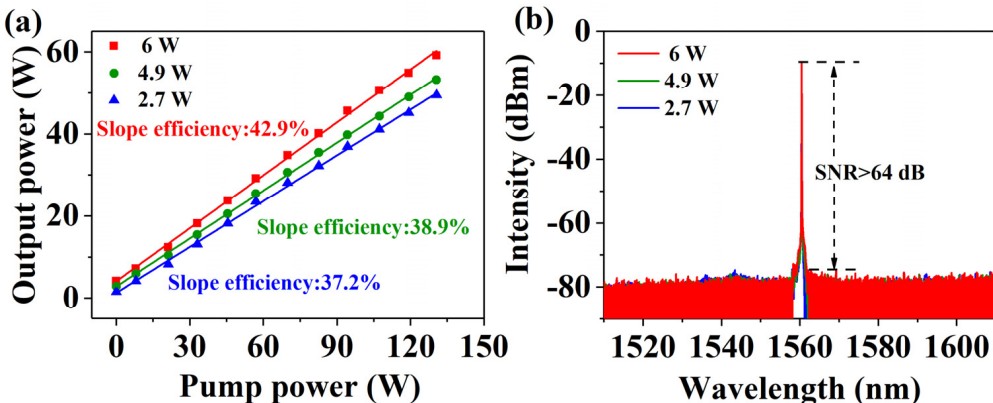

**Figure 3.** The system is under different input signal powers. (**a**) The output power of 1560 nm linearly-polarized SFFL versus the pump power. (**b**) The output spectrum of 1560 nm linearly-polarized SFFL under 40 W output power.

Based on the experimental conclusions obtained above, the pump wavelength and the injection signal light power of the power amplifier are set as 940 nm and 6 W, respectively. The final output power versus the pump power is shown in Figure 4a. The output power

increases linearly without the power saturation effect. Under the pumping power of 252 W, the maximum output laser power is up to 102 W, with the corresponding slope efficiency of 40.5%. Compared with the slope efficiency of 42.9% at a low injection of 940 nm pump power, as shown in Figure 3a, the slope efficiency of 40.5% decreases. This phenomenon can be ascribed to the fact that under high pump power, Yb-ASE becomes stronger due to the Yb–Er energy transfer bottlenecking effects, which limit the amplifier power enhancement, resulting in the slight decline of overall slope efficiency. Meanwhile, the backward power at different pump power is also measured, as displayed in Figure 4a. With the pumping power below 120 W, the backward power gradually increases, closing to a linear trend, while at more than 120 W, the backward power grows rapidly. At the maximum output power, the backward power reaches 526 mW, which is near the threshold of the optical device to withstand the backward power. It is worth noting that this is the main reason for limiting further power enhancement. At the maximum output power of 102 W, the laser spectrum is measured by an optical spectrum analyzer, with a scanning resolution of 0.02 nm. Utilizing a wedge splitter and a focusing mirror, the power injected into the spectrometer is controlled to be 0.5 mW. From Figure 4b, a sharp unimodal distribution with an OSNR of >64 dB is observed. In the range of 1510–1610 nm, there is no obvious ASE. The 1.5 μm high-power SFFL reported in recent years has an OSNR lower than 58 dB, in comparison to the higher OSNR obtained in this 1560 nm SFFL experiment [13,16,17]. Besides, the longitudinal-mode characteristic of the 1560 nm high-power linearly-polarized SFFL is surveyed by a scanning Fabry–Perot interferometer, with a free spectral range (FSR) of 1.5 GHz and a resolution of 7.5 MHz. As shown in the inset of Figure 4b, it can be seen that the final output laser has strict single longitudinal mode characteristics, without mode hop and mode competition phenomena.

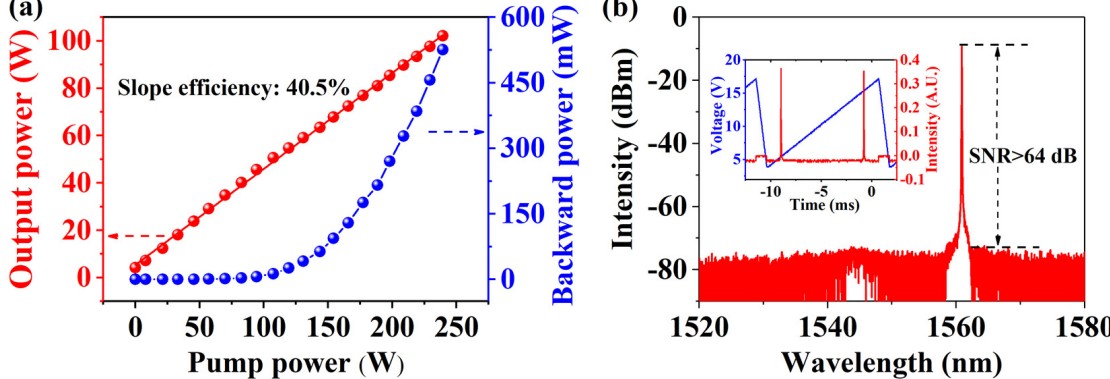

**Figure 4.** The system is under 940 nm LD pumping and 6 W input signal power. (**a**) The output power and the backward power of the 1560 nm high-power linearly-polarized SFFL versus the pump power. (**b**) The optical spectrum of this 1560 nm linearly-polarized SFFL at the maximum output power. Inset: Longitudinal mode characteristic of this 1560 nm linearly-polarized SFFL at maximum output power.

In addition, taking advantage of a polarizing beam splitter (PBS) and two-power-meter, the PER results of 1560 nm linearly-polarized SFFL are measured under different output powers, as displayed in Figure 5a. When the output power is lower than 70 W, the PER maintains 17~18 dB. In contrast, for this laser with higher output power, the PER slowly decreases with the increase in the output power. Finally, when the output power is up to 102 W, the PER is 15.5 dB. The origin of the PER decrease can be attributed to the stress variation from the accumulated heat effects in the PM fiber, especially during the high-power amplification stage. Fortunately, the 1560 nm SFFL has a higher PER compared to the current hectowatt-scale 1.5 μm SFFL, which we attribute to better thermal management of the gain fiber immersed in water [16,17]. The laser power stability at one hour at the maximum output power is measured, as shown in the inset of Figure 5a. The laser average power instability, which is half the value yielded by dividing the maximum

power fluctuation by the average power, is less than ±0.88% at one hour. The power fluctuations can be ascribed to a large amount of heat released when amplified, along with uneven heat dissipation. Further power stability improvements can be achieved through closed-loop feedback.

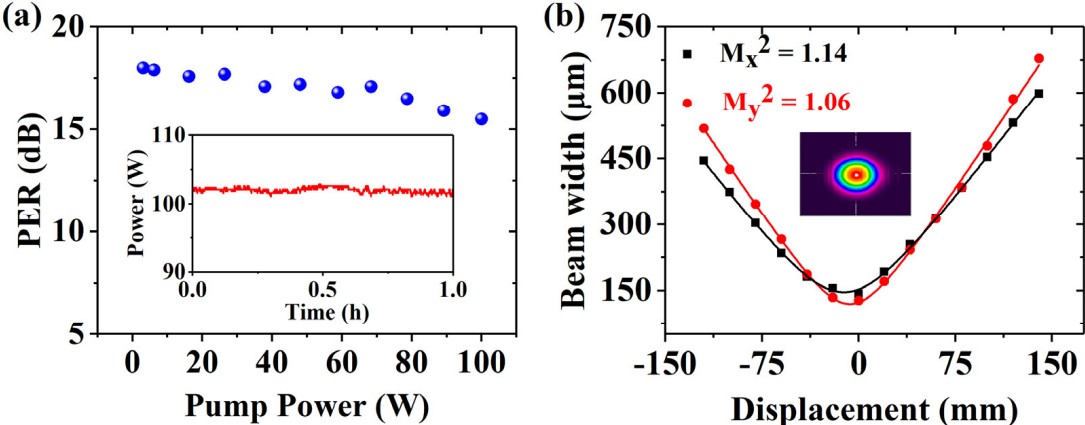

**Figure 5.** The system is under 940 nm LD pumping and 6 W input signal power. (**a**) PER of 1560 nm linearly-polarized SFFL versus the output power. Inset: Power stability at one hour at maximum output power. (**b**) Measured beam quality factors $M^2$ and far-field beam profile at the output power of 102 W.

Using a slit interferometer to test the transverse mode quality of the 1560 nm high-power laser system at the maximum power, based on the hyperbolic fitting method, the spot diameter is measured at different positions of the beam transmission path. The measured $M_x^2$ and $M_y^2$ are 1.14 and 1.06, respectively, as displayed in Figure 5b. The inset result shows the far-field beam profile of the beam quality, which depicts a well quasi-Gaussian spatial beam. Owing to the small inner diameter of the coil and other filtering technology of a high-order transverse mode, the laser has presented a good beam quality, which is close to the diffraction limit [19].

The spectral linewidth test results of the seed laser and three amplifiers are shown in Figure 6a, which is measured by a delayed self-heterodyne method involving a 50 km fiber-delayed Mach–Zehnder interferometer and an 80 MHz fiber-coupled acoustic optical modulator. It can be seen that the four linewidth curves are almost coincident, which reveals that the linewidth is not broadening during the power enhancement process. The test data line is fitted with a Lorentzian curve, and the 20 dB spectrum width is 61 kHz, which means that the 3 dB linewidth of this 1560 nm high-power laser is 3.05 kHz. This favorable result benefitted from the effective control of the ASE in the 1.5 μm band, which can be confirmed by the aforementioned optical spectrum result [13,20]. Compared with the linewidth broadening of each amplification stage that occurs when pumping at 976 nm, the off-peak pumping method of the 1560 nm SFFL effectively suppresses the generation of ASE [13]. The relative intensity noise (RIN) of the output light is measured using an InGaAs photodetector with a 3 dB bandwidth of 150 MHz and an electrical spectrum analyzer, as shown in Figure 6b. It can be seen that the RINs of the laser beams tested at different positions are coincident. The relaxation oscillation peak is located at 0.8 MHz. For frequency ranges greater than 2 MHz, RIN begins to drop rapidly to −130 dB/Hz, with the frequency increasing. To further meet the demand for sensitivity in GWD, the method combining photoelectric feedback with a semiconductor optical amplifier (SOA) can be used to suppress RIN in future work [18,21].

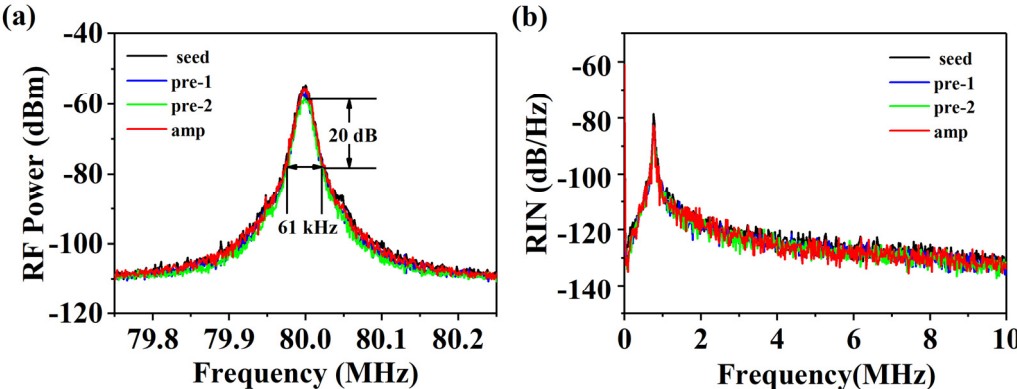

**Figure 6.** The performance test results of seed laser, first pre-amplifier, second pre-amplifier, and power amplifier. (**a**) Spectral linewidth; (**b**) RIN spectrum in 0–10 MHz frequency range.

## 4. Conclusions

In conclusion, we have experimentally demonstrated a 1560 nm high-power narrow-linewidth linearly-polarized SFFL with near diffraction-limited beam quality. By specifically choosing the pumping wavelength and the input signal power in the final power amplifier stage of this laser system, the Yb–Er energy transfer efficiency and the ability of the signal laser to capture pump light are promoted simultaneously, so that the slope efficiency and the output power have been synchronously improved. Finally, under the 940 nm MM-LD pump and the input signal power of 6 W, the highest output power of 102 W with the maximum slope efficiency of 40.5% is obtained. A PER of 15.5 dB, a linewidth of 3.05 kHz, an OSNR of >64 dB, and a near diffraction-limited beam quality of $M_x^2 = 1.14$, $M_y^2 = 1.06$ are obtained simultaneously in this laser system at the maximum output power. The 1560 nm SFFL can further improve the detection range and the detection accuracy of the coherent Doppler lidar, which is benefited from its characteristics of eye safety, high output power, and narrow linewidth. Moreover, with its excellent beam quality and linear polarization characteristics, the 1560 nm SFFL can also help to promote the sensitivity of gravitational wave detection.

**Author Contributions:** Conceptualization, Q.Z. (Qilai Zhao); methodology, S.X., Z.Y. and Q.Z. (Qinyuan Zhang); validation, J.H., C.Y., K.Z. and Z.F.; formal analysis, J.H. and Q.Z. (Qilai Zhao); investigation, J.H., J.Z., Q.G. and W.J.; resources, J.H., Q.Z. (Qilai Zhao) and S.X.; data curation, C.H.; writing—original draft preparation, J.H.; writing—review and editing, Q.Z. (Qilai Zhao); visualization, J.H. All authors have read and agreed to the published version of the manuscript.

**Funding:** This work was supported by the Key-Area Research and Development Program of Guangdong Province (2018B090904001, 2018B090904003, and 2020B090922006), the Major Program of the National Natural Science Foundation of China (61790582), NSFC (62035015), the Fundamental Research Funds for the Central Universities (2020CG03 and 2020ZYGXZR073), the Leading Talents of Science and Technology Innovation of the Guangdong Special Support Plan Program (2019TX05Z344), the China Postdoctoral Science Foundation (2021M101256), Guangdong Basic and Applied Basic Research Foundation (2022A1515012594), the Local Innovative and Research Teams Project of Guangdong Pearl River Talents Program (2017BT01X137), and the Independent Research Project of the State Key Lab of Luminescent Materials and Devices, South China University Of Technology (Skllmd-2022-13).

**Data Availability Statement:** The data presented in this study are available on request from the corresponding author. The data are not publicly available due to privacy concerns.

**Conflicts of Interest:** The authors declare no conflict of interest.

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
