# Peer review of "A 102 W High-Power Linearly-Polarized All-Fiber Single-Frequency Laser at 1560 nm"

_photonics, doi:10.3390/photonics9060396_

Round 1

Reviewer 1 Report

In the manuscript ' A 102 W high-power linearly-polarized all-fiber single-frequency laser at 1560 nm, ' the authors Huang et al. reported their work on the 100-W level single-frequency laser at 1560 nm with the linear polarisation operating. Three-level amplifier was adopt with the spectrum linewidth of 3.05 kHz outputting by effectively controlling the SRS and ASE. This manuscript is suggested to be accept as the publication of Photonics after the following questions are answered.

  1. The author should briefly introduce the method they adopt in optimising the fiber length to reduce both SRS and ASE.
  2. A very key point was missed in the manuscript and that is the cooling system in both pump source and amplifier which will determine the stability of the laser. Especially, did the author apply cooling setups in the fusion positions between 2nd pre-amplifier and main amplifier?
  3. To further clarify the explanation on the ASE effect happened in the experiment, the authors are encouraged to give a energy-level graphic of Er-Yb fiber. However, this is not necessary.

Author Response

Dear Reviewer,

Thank you very much for handling the review of our manuscript. We would like to express our gratitude for your valuable comments and suggestions on improving our manuscript. We have taken all the comments into consideration and have made appropriate changes to the manuscript in order to address them. Our point-by-point responses appear below, in which we first echo your comments and then respond to them.

We hope that these revisions could satisfy your concerns and that they meet the publication requirements. Thank you very much for your attention and consideration to our paper. We are looking forward to your final decision.

Sincerely,

Qilai Zhao, Shanhui Xu, on behalf of all other authors

South China University of Technology, China

Reviewer 2 Report

The paper demonstrated a  1560 nm high-power (around 100W)  linearly-polarized all-fiber single-frequency narrow-linewidth la- 20 ser with near-diffraction-limited beam quality. The paper is technically sound.

It would be very instructive for the readers to include some information about the system's thermal management, especially in the splicing points. In any condition, do the splicing points suffer catastrophic damage? What are the splicing loss in the high power section, and how are their stability? What level of temperature achieve?

Author Response

(The authors gave the same response as above.)

Reviewer 3 Report

The authors of the paper “A 102 W high-power linearly-polarized all-fiber single-frequency laser at 1560 nm” have experimentally demonstrated a 1560 nm high-power narrow-linewidth linearly-polarized single-frequency fiber lasers with near-diffraction-limited beam quality. Under the 940 nm multimode laser diode pump and the input signal power of 6 W, the highest output power of 102 W with the maximum slope efficiency of 40.5% was obtained. A polarization extinction ratio of 15.5 dB, a linewidth of 3.05 kHz, an optical signal-to-noise ratio of >64 dB, and a near-diffraction-limited beam quality of Mx2=1.14, My2=1.06 were obtained simultaneously in this laser system at the maximum output power.

The subject of the paper is useful. It can be published after several changes.

1). I recommend to replace Fig. 2 with two figures: Fig. 2 (devoted to the information concerning the different pumping wavelengths) and Fig. 3 (devoted to the information concerning the different input signal power).

2). All figure captions have to include sentences describing these figures. Only after such sentences descriptions with (a), (b) and so on are acceptable.  

For example,

Figure 2. The system under different pumping wavelengths. (a) The output power of 1560 nm linearly-polarized SFFL versus the pump power. (b) …”

3). In “Conclusions” I recommend to add a sentence describing why the 1560 nm high-power linearly-polarized SFFL is expected to become a reliable light source for the long-range coherent Doppler lidar and the next generation of gravitational wave detection.

Author Response

(The authors gave the same response as above.)
